# Farnesol, a Quorum-Sensing Molecule of *Candida albicans* Triggers the Release of Neutrophil Extracellular Traps

**DOI:** 10.3390/cells8121611

**Published:** 2019-12-11

**Authors:** Marcin Zawrotniak, Karolina Wojtalik, Maria Rapala-Kozik

**Affiliations:** Department of Comparative Biochemistry and Bioanalytics, Faculty of Biochemistry, Biophysics and Biotechnology, Jagiellonian University in Krakow, 7 Gronostajowa st., 30387 Krakow, Poland; marcin.zawrotniak@uj.edu.pl (M.Z.); karolinawojtalik13@gmail.com (K.W.)

**Keywords:** neutrophil extracellular traps, NETs, *Candida albicans*, quorum sensing, farnesol

## Abstract

The efficient growth of pathogenic bacteria and fungi in the host organism is possible due to the formation of microbial biofilms that cover the host tissues. Biofilms provide optimal local environmental conditions for fungal cell growth and increased their protection against the immune system. A common biofilm-forming fungus—*Candida albicans*—uses the quorum sensing (QS) mechanism in the cell-to-cell communication, which determines the biofilm development and, in consequence, host colonization. In the presented work, we focused on the ability of neutrophils—the main cells of the host’s immune system to recognize quorum sensing molecules (QSMs) produced by *C. albicans*, especially farnesol (FOH), farnesoic acid (FA), and tyrosol (TR), with emphasis on the neutrophil extracellular traps (NETs) formation in a process called netosis. Our results showed for the first time that only farnesol but not farnesolic acid or tyrosol is capable of activating the NET production. By using selective inhibitors of the NET signaling pathway and analyzing the activity of selected enzymes such as Protein Kinase C (PKC), ERK1/2, and NADPH oxidase, we showed that the Mac−1 and TLR2 receptors are responsible for FOH recognizing and activating the reactive oxygen species (ROS)-dependent netosis pathway.

## 1. Introduction

Pathogenic microorganisms such as bacteria and fungi are able to successfully attack and colonize the host organism only while living in colonies. Colonies ensure the maintenance of appropriate local environmental conditions, increase local concentrations of many released compounds, such as proteolytic enzymes, and enhance the resistance to the host immune cells [1]. The development of infection is associated with the creation of biofilms - microbial assemblies that cover significant areas of host tissue. A critical factor for the optimal existence of microorganisms within biofilms is proper communication between pathogenic cells. It ensures synchronization of morphological changes, growth, gene expression and secretion of many compounds. The cell-to-cell communication system, called quorum sensing (QS) involves numerous quorum sensing molecules (QSMs), whose presence and concentration in the biofilm regulate biofilm functioning [2]. The first fungal microorganism in which the QSM mechanisms were identified was *Candida albicans*—an opportunistic yeast-like pathogen that resides on the skin and mucous membranes, which, due to a wide range of virulence factors, is responsible for the development of serious and hardly curable infections—candidiases [3,4,5,6]. In the properly functioning *C. albicans* QS, the autoregulatory QSMs are involved, including the best known farnesol (FOH) [3], farnesoic acid (FA) [7], and tyrosol (TR) [8,9]. In addition, *C. albicans* secretes phenylethanol and tryptophol, however, whether these two aromatic alcohols can act as QSMs in this fungus still remains to be established [5]. FOH is a sesquiterpene alcohol made up of three isoprene units that is secreted by *C. albicans* into the extracellular space, reaching a concentration of over 50 µM, but the local QSM concentrations may be significantly higher [10,11]. *C. albicans* ATCC 10,231 strain was reported to secrete FA instead of FOH [12]. The presence of QSMs regulates the expression of many genes, including those responsible for the production of yeast virulence factors [13,14]. FOH was shown to work in an autoregulatory fashion, inhibiting the transition of *C. albicans* from the yeast-like to filamentous forms [3], thus blocking the formation of biofilm [15,16]. Its action is contrary to the second QSM-TR, a tyrosine-related alcohol that stimulates hypha production during the early stages of biofilm development [8]. FOH also affects the expression of genes responsible for the protection of *Candida* spp. against oxidative stress [17,18]. Moreover, FOH is used by fungi in the coexistence with *Pseudomonas aeruginosa* where this QSM down-regulates quinolone production by bacteria, thus enabling the coexistence of these two species [19]. Upon contact with the host, FOH presents immunomodulatory properties [20], affecting the efficiency of macrophages by decreasing their phagocytic activity [21], with the stimulation of the inflammatory cytokine expression [22]. FOH is also involved in blocking of monocyte differentiation into immature dendritic cells (DCs) and modulation of the DC’s ability to induce T cell proliferation and activation of neutrophils [23].

Neutrophils (PMNs) can identify *C. albicans* and respond very quickly to the appearance of fungal cells by phagocytosis or the release of structures called neutrophil extracellular traps (NETs) [24,25]. However, the mechanism used by neutrophils to select between these two processes is still unknown. NETs are built of DNA backbone decorated with granular proteins like elastase, cathepsin G, proteinase 3 and myeloperoxidase (MPO), responsible for effective killing of pathogens. The process of NET release is also a mechanism of cell death, which results in the rupture of the cell membrane and the release of cellular content into the extracellular space [26]. In contrast, phagocytosis uses membrane tubulovesicular extensions (cytonemes) to capture pathogens without the neutrophil’s death [27]. These components of NETs allow to defend against the hyphal form of *C. albicans* cells, which due to their size, cannot be effectively phagocyted [28]. *C. albicans* cells are trapped within the NET structures and then killed by granular enzymes and reactive oxygen species [26,29]. Many studies have indicated that some of *C. albicans’* virulence factors can activate NET release, the process named netosis [28,29,30]. Among them are glucans and mannans—the components of the fungal cell wall, as well as secreted aspartyl proteases (Saps), all of which can stimulate the netosis. However, the studies showing that the number of NETs significantly increases upon contact with the filamentous form of the pathogen, indicated that the morphology of fungal cells and, consequently, their size can determine the type of neutrophil response [28,30].

Although the influence of FOH on neutrophils has been demonstrated [23] there is no information about the possible netosis activation by QSMs, especially QSMs released by *C. albicans*. Therefore, the aim of this study was to determine the potential of FOH, FA, and TR to trigger NET release and to find the netosis signaling pathway activated by QSMs, as well as to verify their chemotactic properties.

## 2. Materials and Methods

### 2.1. Isolation of Neutrophils

Human neutrophils were isolated from EDTA-treated whole blood delivered by the Regional Blood Donation Center, Kraków, Poland, obtained from healthy donors. The neutrophil-containing fraction was isolated by Pancoll gradient separation [30]. The fraction containing neutrophils and erythrocytes was mixed with a solution of polyvinyl alcohol (1%) and incubated for 20 min at room temperature. The upper layer was collected and centrifuged. Erythrocytes were removed by lysis in a hypotonic solution. Cells were washed and resuspended in phosphate buffered saline (PBS). The neutrophil purity was typically >95%, as assessed by forward-scatter and side-scatter flow cytometric analyses.

### 2.2. Viability Assay

#### 2.2.1. Caspase 3/7 Activity

The cell apoptosis was analyzed by measuring the activity of proapoptotic caspases 3/7. Neutrophils (2 × 10^5^ cells/well), suspended in solutions of FOH (trans,trans−3,7,11-Trimethyl−2,6,10-dodecatrien−1-ol; Sigma-Aldrich, St. Louis, MO, USA), FA (Echelon Biosciences Inc, Salt Lake City, UT, USA) or TR (Sigma-Aldrich, St. Louis, MO, USA) at variable concentrations, were placed in the wells of a 96-well white microplate and incubated for 1 h at 37 °C, at 5% CO_2_. Then, the cells were washed with PBS, and 100 µL of Caspase-Glo^®^ 3/7 Reagent (Caspase-Glo^®^ 3/7 Assay, Promega, Madison, WI, USA) was added to each well, the plate was gently mixed by shaking at 300 rpm for 30 s and the chemiluminescence was measured continuously for 2 h at 37 °C.

#### 2.2.2. Annexin V/Propidium Iodide Analysis (Flow Cytometry)

The disturbance of the cell membrane was monitored using Annexin V (AnV) and propidium iodide (PI). Neutrophils (1 × 10^6^ cells/sample) were placed in an eppendorf tube, washed twice with PBS, and resuspended in solutions of FOH or FA at variable concentrations. Unstimulated cells served as a negative control, and phorbol 12-myristate 13-acetate (PMA)-treated neutrophils represented a positive control. Cells were incubated for 1 h at 37 °C, at 5% CO_2_, washed three times with PBS, and stained with propidium iodide and fluorescein isothiocyanate (FITC)-labeled Annexin V for 15 min, according to supplier’s instruction (Dead Cell Apoptosis Kit with Annexin V-FITC and PI, Invitrogen, Carlsbad, CA, USA). Then, cells were analyzed with flow cytometry (LSR Fortressa, BD, San Jose, CA, USA).

### 2.3. Analysis of ROS Production

The production of reactive oxygen species (ROS) by neutrophils was analyzed using chemiluminescence measurements. Neutrophils (2 × 10^5^ cells/well) were suspended in 160 µl of Krebs-Ringer phosphate buffer containing freshly prepared luminol solution (10^–6^ M) and allowed to settle for 15 min at 37 °C, 5% CO_2_ in the wells of 96-well white microplate. Then, 10 µl of FOH, FA, or TR were added at variable concentrations. Untreated neutrophils were used as a negative control, and cells stimulated with 25 nM PMA as a positive control. The chemiluminescence of luminol was recorded for one hour with one-second integration time, using a BioTek Synergy H1 microplate reader.

### 2.4. Analysis of NET Quantity and Image

Neutrophils (2.2 × 10^5^ cells/well) were seeded into the well of 96-wells black microplate in 150 µL of RPMI-1040 and allowed to settle for 15 min at 37 °C, 5% CO_2_. Then, neutrophils were stimulated with 150 µL of FOH, FA or TR at variable concentrations in RPMI-1040. Unstimulated cells and cells treated with 25 nM of PMA served as negative and positive controls, respectively. Plates were incubated at 37 °C, 5% CO_2_ for 3 h and then analyzed, as follows:

For the visualization of NETs: the samples were fixed with 3.8% paraformaldehyde for 15 min and washed with PBS. Sytox Green (Thermo Fisher, Waltham, MA, USA.) dye was added to each well at the final concentration of 1 µM to visualize extracellular DNA. Imaging was performed using a Nikon Eclipse Ti microscope (Nikon Instruments, Melville, NY, USA).

For extracellular DNA quantification: the cells were washed three times with PBS and then 50 µL micrococcal nuclease (MNase, 1 U/mL) was added to cleave and release small fragments of extracellular DNA, and the microplates were incubated at 37 °C for further 20 min. The enzymatic reaction was stopped by the addition of EDTA solution (100 µg/mL), and after cell centrifugation (350× *g*, 5 min) 50 µL of supernatant was transferred into 96-well black microplate containing Sytox Green at a final concentration of 1 µM. Fluorescence was measured using the Biotek Synergy H1 microplate reader at the excitation wavelength of 465 nm and the emission wavelength of 525 nm.

### 2.5. Identification and Quantification of Myeloperoxidase

Neutrophils (2.2 × 10^5^ cells/well) were seeded into the well of 96-wells black microplate in 150 µL of RPMI-1040 and allowed to settle for 15 min at 37 °C, 5% CO_2_. Then, neutrophils were stimulated with 150 µL of FOH at a concentration of 100 µM and 200 µM.

For the visualization of MPO: the samples were fixed with 3.8% paraformaldehyde for 15 min and washed with PBS. 50 µL of 1:100 diluted primary mouse anti-MPO antibodies (Abcam, Cambridge, UK) was added to each well and incubated for a night at 4 °C. Then, cells were washed three times with PBS and incubated with 1:500 diluted secondary Alexa Fluor 555 anti-mouse antibodies (Abcam, Cambridge, UK) for 1 h at 37 °C. Cells were washed two times with PBS, and imaging was performed using a Nikon Eclipse Ti microscope.

For the quantification of MPO: the quantitative determination of MPO was performed using Human MPO ELISA Kit (Wuhan Fine Biotech Co., Ltd., Wuhan, China). The cells were washed three times with PBS and then 50 µL micrococcal nuclease (MNase, 1 U/mL) was added to release DNA-bounded proteins, and the microplates were incubated at 37 °C for further 20 min. After cell centrifugation (350× *g*, 5 min) 100 µL of supernatant was transferred into anti-MPO pre-coated wells of plate, and the manufacturer’s instruction was followed.

### 2.6. Recognition of Receptors Involved in Netosis

Neutrophils were preincubated with specific antibodies or inhibitors prior to stimulation with *C. albicans* factors. Neutrophils (1 × 10^6^) were preincubated for 30 min at 37 °C in RPMI-1640 medium with 1 µg/mL of blocking antibodies directed against TLR2, TLR4 (Invivogen, Toulouse, France), CD11a, CD11b, CD16, CD18 (BioLegend, San Diego, CA, USA) or isotype control antibody—IgG (Abcam, Cambridge, UK).

### 2.7. Analysis of Protein Kinase C (PKC)

The activity of PKC was monitored using PepTag^®^ Non-Radioactive Protein Kinase Assays (Promega, Madison, WI, USA). Neutrophils (5 × 10^6^ in 500 µL of PBS per well) were stimulated with 100 µM and 200 µM FOH in 12-well microplate. Negative and positive controls were prepared as described above. After 1 h of incubation at 37 °C, at 5% CO_2_, cells were washed with PBS, resuspended in 500 µL of cold PKC extraction buffer, and homogenized in the cold. Lysates were centrifugated for 5 min at 4 °C, 14,000× *g* and supernatants were purified on diethylaminoethyl (DEAE) cellulose resin. To the reaction solution containing 5 µL of PepTag^®^ PKC Reaction Buffer, 2 µg of PepTag^®^ C1 Peptide, and 5 µL of sonicated PKC Activator, 10 µL of purified samples or 4 µL of Protein Kinase C at concentration of 2.5 µg/mL (as a positive control) were added, followed by incubation at 30 °C for 30 min. Then, the reaction was stopped by placing the tube in a 95 °C heating block for 10 min. 1 µL of 80% glycerol was added to each sample, and then samples were electrophoretically separated on a 0.8% agarose gel at 100 V for 15 min. The phosphorylated peptide migrated to the cathode (+), while the non-phosphorylated peptide migrated to the anode (-). The gel was photographed on a transilluminator.

### 2.8. Analysis of ERK1/2

The amount of total and phosphorylated ERK1/2 was quantified using SimpleStep ELISA Kit (Abcam, Cambridge, UK). Neutrophils (1 × 10^6^ in 500 µL of PBS per well) were stimulated with 100 µM and 200 µM FOH in 12-well microplate. Negative and positive controls were prepared as described above. After 1 h of incubation at 37 °C in 5% CO_2_, cells were lysed using a Cell Extraction Buffer (Abcam, Cambridge, UK). The protein concentration in the lysate was determined using the Bradford assay [31]. Then, 50 µL of lysate was mixed with 50 µL of antibody cocktail (anti-ERK1/2–total or anti-pT202/Y204–phosphorylated ERK1/2 provided by the manufacturer) in wells of SimpleStep pre-coated 96-well microplate, according to manufacturer’s instruction. After 1 h of incubation with gentle shaking at room temperature, the wells were washed 3 times with PBS and then TMB (3,3′,5,5′-tetramethylbenzidine) solution was added. After 15 min, the reaction was stopped with a Stop solution, and absorbance at 450 nm was recorded using a BioTek Synergy H1 microplate reader.

### 2.9. Analysis of Netosis Signaling Pathway

For the analysis of netosis signaling pathway, selected enzyme inhibitors were used. Neutrophils were pre-treated for 30 min prior to stimulation with different inhibitors: 30 µM piceatannol (Syk inhibitor; Sigma-Aldrich, St. Louis, MO, USA), 10 µM PP2 (Src inhibitor; Calbiochem, Darmstadt, Germany), 10 µM UO126 (ERK inhibitor; Cell Signaling Technology, Beverly, MA, USA) or 5 µM DPI (NADPH oxidase inhibitor; Sigma-Aldrich, St. Louis, MO, USA). Then, the cells were stimulated and analyzed as described above.

### 2.10. Analysis of Neutrophil Chemotaxis

Neutrophil migration was evaluated by the 24-well microchemotaxis chamber technique (Transwell^®^-Clear inserts, Corning, NY, USA). Neutrophils were labeled for 10 min with 1 µM CellTracker Red solution (Invitrogen, Carlsbad, CA, USA), washed three times with PBS, and placed in the upper compartment of the chamber (1 × 10^6^ cells/well). Samples of FOH (various concentrations of up to 400 µM) or fMLP (1 µM; used as a positive control) were placed in the lower compartment. PBS was used as a negative control. The compartments were separated by a membrane with 3 µm pores. The chambers were incubated at 37 °C in 5% CO_2_ atmosphere for 1 h. Neutrophil migration was monitored by fluorescence microscopy (Motic AE31E, MoticEurope, Barcelona, Spain), and the number of cells migrated into the lower compartment of the chamber was quantified.

### 2.11. Statistical Analysis

Each of the experiments was repeated at least three times, obtaining consistent results. Two replicates were performed in each experiment. The graphs show the results of a single representative experiment.

Statistical analysis was performed with the GraphPad Prism 7 software (GraphPad Software, CA, USA). The statistical significance was assessed by ANOVA and Dunnett’s multiple comparisons post-test.

## 3. Results

### 3.1. Farnesol but Not Farnesoic Acid or Tyrosol Triggers NET Formation

Neutrophils are the cells with high microbiocidal potential, and their high killing efficiency is due to their ability to recognize many “foreign” molecules released by pathogens. Owing to these properties and because their active cells move under the chemoattractant gradient, neutrophils release their antimicrobial molecules directly at the site of infection. In our current study, we focused on the role of *C. albicans*-released QSMs, in particular FOH.

We verified the neutrophil responses to contact with QSMs released by *C. albicans,* focusing on the PMN ability to release NETs. The stimulation of PMNs with FOH showed the dose-dependent responses with NET release within the whole range of concentrations tested (Figure 1a). The highest level of NETs released by FOH-treated neutrophils was observed for 250 µM FOH, and it reached about 60% of the positive control.

Different results were obtained for FA, a farnesol derivate. PMNs did not release NETs in response to any of the examined concentrations of FA. Although at the concentration of 250 µM the fluorescent signal increased significantly, it was probably a result of cell death induced by this FA concentration. Similar results were observed for TR, the third examined QSM. No released NETs were identified within the entire range of examined TR concentrations.

These quantitative results were confirmed by fluorescence microscopy analysis (Figure 1b), on which Sytox Green-stained neutrophils, stimulated with FOH showed cloud-like structures located around the human cells and composed of green-labeled DNA.

To verify the NETs production during FOH treatment, also the extracellular localization of myeloperoxidase was determined, using specific, fluorescent antibodies. During netosis, MPO is moved from granule to the nucleus and then released together with DNA in the form of NETs.

Quantitative analysis of MPO in the samples containing neutrophils treated with FOH, was performed using ELISA. For this purpose, the netting cells were prior washed with PBS, and the extracellular DNA was digested with MNase to liberate bound MPO. In Figure 2a, we showed MPO concentration in the cell supernatants, determined by ELISA. The amount of detected MPO correlated with the DNA quantity, confirming that the observed DNA structures belong to the NETs. Additionally, MPO/DNA complexes were visualized microscopically (Figure 2b). As was shown in the figure, the protein location correlated with DNA clouds released by PMNs, confirming activation of netosis by FOH.

### 3.2. Farnesol Treatment of Neutrophils Does Not Lead to Cell Apoptosis

In order to confirm that the observed effect of the DNA release the cell was associated with the mechanism of netosis, but not cell death like apoptosis, the activation of proapoptotic caspases was analyzed. PMNs were exposed to FOH and FA at the concentration range of 0–1200 µM for one hour, and caspase 3 and 7 activities were measured using the chemiluminometric method (Figure 3).

The results presented in Figure 3 indicated that FOH at the concentration range of 0–200 µM did not activate proapoptotic caspases in neutrophils. However, higher concentrations of FOH led to apoptotic cell death. FA-treated neutrophils showed activation of apoptosis in a dose-dependent manner within the whole tested range of concentrations. This explains the absence of NETs released in the presence of FA, because activation of caspases 3 and 7 inhibits the release of NETs [32].

Besides, the neutrophils were incubated with FOH and FA at concentrations of 100 µM and 200 µM for one hour and then labeled with PI and AnV. The cell viability analysis based on PI and AnV-FITC was performed using flow cytometry. The results (Figure 4a), comparison to the 3/7 caspase analysis, also allow assessing the potential of PMNs for the release of NETs. The changes in the distribution of cell population labeled with PI and AnV (Figure 4b), characteristic to netosis, were previously described and used by Masuda et al. [33].

The results showed four groups of cells: AnV^−^PI^−^ identified as live; AnV^+^PI^−^ identified as early apoptotic; AnV^+^PI^+^ identified as apoptotic; AnV^−^PI^+^ identified as necrotic. However, the obtained cell distribution patterns, especially for AnV^+^PI^+^ cells, were identical for cells in the netosis, represented by PMNs stimulated with PMA. Approximately 35%–40% of FOH-treated neutrophils showed positive labeling for both AnV and PI, similarly to the PMA-treated positive control. This indicated that these cells lost the integrity of the cell membrane, however, the comparison of these results with the activation of caspases led to the conclusion that observed changes did not result from apoptotic cell death. The low percentage of PI-positive neutrophils suggests that DNA released outside the cells was not due to mechanical destruction or necrotic death. The presence of approximately 15% of AnV^+^PI^−^ cells suggests that neutrophils were in the ongoing netosis process. In contrast, FA-treated PMNs did not show apoptotic and necrotic cell traits within the tested range of QSM concentrations, within one hour of contact with FA.

### 3.3. Farnesol Induces Rapid ROS Production

The activation of NADPH oxidase, resulting in the production of ROS is a key step in the ROS-dependent netosis pathway. The chemical stimulation of neutrophils with PMA leads to the release of a high amount of ROS within minutes of activation. We checked the ability of the examined QSMs to activate NADPH oxidase and release ROS. PMNs were treated with selected doses of FOH or FA, and the activity of NADPH oxidase was measured using a chemiluminescence-based assay.

Changes in chemiluminescence intensity over the time showed in Figure 5 were proportional to the amount of ROS produced by PMNs. The stimulation of neutrophils with FOH at the concentration of 100 and 200 µM caused the rapid activation of NADPH oxidase and release of ROS at a time similar to PMA-stimulated cells. The level of ROS released by PMNs treated with 100 µM FOH reached about 25% of the positive control response, while activation with 200 µM FOH resulted in two-fold higher ROS production. Release of ROS by neutrophils correlated with the production of NETs, suggesting that FOH activates the ROS-dependent mechanism of netosis.

The response of neutrophils to FA presented relatively low chemiluminescence signal, confirming that this QSM does not participate in NET release.

### 3.4. Mac-1 and TLR2 Surface Neutrophil Receptors Are Involved in Farnesol-Induced Activation of Netosis

Many of the neutrophil surface receptors are involved in the recognition of molecules derived from pathogenic microorganisms, which leads to activation of the netosis signaling pathway and the NET release. We selected six receptors known to be able to mediate the release of NETs and checked their role in FOH-induced netosis. The receptors were blocked with specific antibodies, and the neutrophils were stimulated with selected concentrations of FOH for three hours. A decrease in the level of fluorescence of Sytox Green-stained DNA suggests the role of this receptor in the activation of netosis. Among selected receptors, we identified that the inhibition of CD11b and CD18 caused a ca. 2.5-fold decrease in the level of released NETs (Figure 6). This result strongly suggests the important role of these receptors in the recognition of FOH and activation of netosis. Moreover, inhibition of the TLR2 receptor caused a reduction of the level of NETs to about 50% of PMA-treated response, also indicating the participation of this receptor in netosis. A small contribution to the activation of FOH-induced netosis could also be assigned to CD11a receptor whose blockade caused a 20% decrease in the amount of released DNA. Other receptors tested—CD16 and TLR4—did not seem to be essential for FOH-induced netosis.

### 3.5. Farnesol Leads to the Activation of Protein Kinase C and ERK1/2

PKC plays a crucial role in the netosis signal pathway. We checked the level of PKC activation in lysates of neutrophils previously treated with two concentrations of FOH or FA for one hour. After pre-purification of the lysates on DEAE cellulose, the kinase activity was determined based on the phosphorylation of the synthetic peptide, which was then subjected to electrophoretic separation. Figure 7a presents an electrophoretic separation of a synthetic peptide whose amount in the phosphorylated form determines the level of PKC activation. Figure 7b shows quantitative PKC activity in assayed samples corresponding to the densitometrically analyzed bands of the phosphorylated form of the peptide on an electrophoretic gel.

The results showed that PKC activity in neutrophils treated with FOH increased rapidly, reaching a two-fold higher level than in unstimulated cells, comparable to that obtained for chemically induced (PMA) activation of this enzyme. The PKC activity leads to the activation of subsequent netosis mediators. On the other hand, PMNs treated with FA did not show any changes in PKC activity. The lack of PKC activation by FA confirms that neutrophils do not release NETs in the presence of this QSM.

The results of the analysis of ERK1/2 kinase activation in neutrophils treated with FOH are shown on Figure 8. The activation of ERK1/2 by 100 µM FOH is two-fold higher than in unstimulated cells, but 200 µM FOH caused three-fold greater phosphorylation of this enzyme. These findings suggest that the activation of the signaling pathway by FOH in neutrophils involving ERK1/2 in a dose-dependent manner.

### 3.6. The ROS-Dependent Netosis Pathway is Activated by Farnesol

Stimulation of surface receptors involved in the induction of netosis leads to the activation of selected mediators of the signaling pathway. We checked the role of the five primary mediators involved in ROS-dependent and ROS-independent netosis pathways. For this purpose, selected mediators were blocked in PMNs using specific inhibitors before neutrophil activation with 200 µM FOH. Fluorescence of Sytox Green-stained extracellular DNA was used to determine NET release.

The results (Figure 9) indicate that each of the tested mediators was involved in FOH-induced netosis, but with a relative variable contribution. Syk and Src kinases co-operated with neutrophil surface receptors, and their involvement in netosis activation appeared to be significant. The inhibition of these proteins caused a 50% decrease in the amount of released NETs. In turn, the role of PI3K in FOH-induced netosis seemed to be less important. PI3K probably played a significant role in the ROS-independent netosis signaling pathway. In our experiments, blockade of this kinase caused a decrease in the amount of extracellular DNA by 20% relative to the control.

ERK1/2 and NADPH oxidase also participate in the activation of the ROS-dependent netosis pathway. The release of NETs depending on ROS production appears to be the primary netosis mechanism involved in neutrophil responses to FOH, as the FOH stimulation of PMNs with blocked ERK1/2 resulted in a 50% reduction of the amount of released NETs. Moreover, NADPH oxidase, which is responsible for the production of ROS in the cells, seems to play the most significant role in the netosis process induced with FOH. Only 20% of DNA was released by neutrophils with the blocked activity of NADPH oxidase compared to the control cells. This result and data showing increased ROS production by neutrophils treated with farnesol confirm that FOH-induced netosis is a ROS-dependent process.

### 3.7. Farnesol Is a Chemoattractant for Neutrophils

Additionally, we showed that FOH may be a chemoattractant for PMNs. To support this hypothesis, the neutrophils were placed in chambers to measure chemotactic ability, and then the chambers were transferred to the FOH solutions at the concentration within a range of 50 to 400 µM. As the negative control, PBS was used, whereas the positive control was represented by the responses of neutrophils to 1 µM solution of fMLP. The number of cells that passed through the membrane according to the factor gradient was counted after one hour of treatment. The obtained results (Figure 10) showed that FOH is recognized by neutrophils as a chemoattractant, causing the PMN movement toward the concentration gradient. The number of cells that passed through the membrane was proportional to FOH concentration in the range of 50 to 200 µM, that corresponds to the concentration of FOH detected under in vivo conditions [23]. The higher concentrations of FOH did not cause any further increase in neutrophil migration, but anyway the maximal level of neutrophil chemotaxis upon FOH treatment reached 60% of the response to fMLP, the positive control. These results suggest that neutrophils can migrate to the *Candida* infection sites in response to QSMs released by the yeast.

## 4. Discussion

Quorum sensing molecules are essential virulence factors of many pathogenic microorganisms-bacteria as well as fungi. The release of QSMs ensures the fast local communication between cells in the infected area, growth synchronization, as well as response to environmental changes of temperature and pH, the presence of biocidal compounds, etc. [34,35].

*C. albicans* yeast mainly use QSMs to regulate growth, change of morphological form and also create biofilms [36]. However, these molecules have an impact on many host cells, like immune cells, such as macrophages, DCs, and neutrophils [20,23].

Our results presented in this paper indicated an important role of neutrophils in the response to QSMs produced by *C. albicans*, which is based on the release of NETs. We showed that farnesol, one of the three compounds identified as *C. albicans* QSMs (FOH, FA and TR) was responsible for activating the netosis, a finding that had not been presented in the literature previously.

The role of FOH released by *C. albicans* is still not fully understood, but it is known that the presence of this compound inhibits the growth of fungal biofilm, and also blocks the change in the morphological transition of fungal cells from blastospores to filamentous forms. Also, high concentrations of FOH can stimulate the yeast to reverse switch from invasive filamentous hyphae back to blastospore [37]. As we presented in the current work, FOH produced at different local concentrations at the site of infection caused the neutrophil response involving NET release. The finding is important for the host defense against *C. albicans* cells because current knowledge has suggested that the morphological form of the fungus determines the mechanism of neutrophil response [28,30]. Some findings indicated that the size of the fungal cells determines the release of NETs, suggesting that the large, filamentous form of *C. albicans* is responsible for activating the netosis [28]. Other studies indicated that the composition of the cell wall of the filamentous form, as well as the released proteolytic enzymes are crucial for the stimulation of neutrophils to NET production [30,38]. Regardless of the identified NET triggers, all studies pointed that *C. albicans* blastospores generate much smaller amounts of NETs, suggesting that this morphological form of *C. albicans* cells can “hide” from neutrophils and avoid the killing. Perhaps, fungi use FOH to inhibit their cell filamentation and progress of infection, just to survive in milieu infiltrated by neutrophils This situation is dangerous for the host because the ‘invisible’ to neutrophils intruder cells can survive and under favorable conditions, develop the difficult to treat, secondary infections [39]. Therefore, the ability of neutrophils to recognize FOH, the essential QSM molecule of the fungus, and release NETs in the response to its production at the place of infection, can significantly affect the effectiveness of host defense against *C. albicans* presence. We confirmed the recognition of FOH by neutrophils using chemotaxis analysis, showing that FOH is an efficient chemotactic factor for neutrophils. To date, there is no literature data showing neutrophil migration caused by FOH. However, FOH is known to be a chemoattractant for macrophages [40].

Moreover, given that the microbiocidal activity of NETs covers a certain area of infection, a direct identification of fungal cells by neutrophils is not required, and the presence of FOH may be sufficient.

Studies performed by Leonhardt et al. regarding the effect of FOH on neutrophils did not show any increasing ability of these host cells to phagocyte and kill *C. albicans* cells. However, the viability of *C. albicans* was analyzed after 1 h of contact, while the classic ROS-dependent netosis pathway leads to NET release after about 2–3 h of neutrophil activation [23]. The short time period used in Leonhard’s analysis may explain the lack of changes in the fungal cell vitality. The same studies showed an increased FOH concentration-dependent release of elastase, myeloperoxidase, and lactoferrin–granule proteins identified in NETs [23,29]. Thus, our observations of NET release may explain the presence of granule proteins upon neutrophil treatment with FOH in Leonhardt’s studies.

Activation of netosis by QSMs has been described in only one case—the *Pseudomonas aeruginosa* bacterium. It was shown that mutations in the quorum-sensing regulatory gene lasR affect the amount and structure of released NETs. In addition, stimulation of neutrophils with purified protein, recombinant endotoxin-free LasA induced NETs in a concentration-dependent manner [41].

The other two analyzed QSMs-FA and TR–did not activate netosis. There are also no literature reports on the impact of these compounds on the functioning of neutrophils. Only TR was identified as a protective agent against phagocytic killing by neutrophils, however, the mechanism of its action is still unknown. One study showed the antioxidant activity of TR [42], which could influence the neutrophil response by blocking ROS production, required for proper functioning of netosis mechanisms. However, other studies showed no antioxidant activity of TR, thus leaving the mechanism of its action on neutrophils unrecognized [18].

The production of extracellular DNA but not apoptosis was demonstrated in this work for neutrophils treated with FOH at the range of concentrations observed locally at the site of *C. albicans* infection. This result is consistent with the observation of Leonhardt et al. [23]. In contrary, we presented that FA is active inducer of neutrophil apoptosis in a dose-dependent manner, the finding that can explain the lack of netosis in neutrophil response to this QSM. These results were confirmed by the cytometric analysis showing the morphological changes of neutrophil cells observed during the FOH-induced netosis process but absent upon treatment with FA.

During the contact of *C. albicans* cells with neutrophils, resulting in NET production, the more common ROS-dependent mechanism of netosis was adopted. The analysis of ROS production by PMNs stimulated with FOH showed that neutrophils release ROS in just a few minutes after activation confirming the ROS-dependent netosis pathway used by neutrophils in response to FOH. In addition, Gilbert et al., identified farnesyl thiotriazole (FTT), a FOH precursor as an activator of NADPH oxidase in neutrophils [43]. However, the production of ROS in response to FA and TR was not observed, confirming a lack of response in the netosis way.

Knowing that FOH is a chemoattractant for neutrophils, and also activates the ROS-dependent netosis pathway, the role of selected surface receptors in activation of netosis by this QSM was analyzed. The results pointed the CD11b/CD18 and TLR2 receptors as being involved in the recognition of FOH and further activation of neutrophils. The CD11b/CD18 receptor engagement in neutrophil chemotaxis and netosis responses to fungal infection was previously demonstrated [30]. Also Leonhardt et al. observed that during stimulation of PMNs by FOH, the level of CD11b receptor on the cell surface increased, while the amount of CD16 receptor decreased [23]. However, Ghosh et al. showed increased TLR2 and Dectin-1 expression in FOH-treated macrophages [22]. No analogous analysis was performed for neutrophils, however, the effect of FOH on TLR2 may also be significant in PMNs.

Further, the role of known mediators of netosis signaling pathway was verified upon neutrophil treatment with FOH. We showed FOH-induced phosphorylation of PKC, an important signal transducer of netosis. Activation of PKC results in the activation of NADPH oxidase and the production of significant amounts of ROS [44]. This result confirms and explains the role of ROS produced by neutrophils in contact with FOH. There is no literature data on the activation of PKC by FOH in neutrophils; however, in human acute leukemia CEM-C1 cells, the synthesis of diacylglycerol (DAG), a PKC activator, was observed in response to 20 µM FOH [45]. In addition, FTT seems to be also a PKC activator [43].

Inhibition of netosis pathway mediators such as Syk, Src, PI3K, and ERK1/2 caused a decrease in the amount of released NETs. Previously, only the pro-apoptotic activity of FOH was known, leading to reduction of PI3K expression in cells and ERK1/2 in HeLa and DU145 prostate cancer cells [46,47]. In turn, Joo et al. also pointed to the pro-apoptotic effect of FOH on cells, however, by activation of ERK1/2 [48]. However, our results did not show the inhibition of these mediators. In contrary, ERK1/2 kinase activity showed a two-fold increase at the activation of the netosis mechanism during which the apoptosis pathway in the PMNs was blocked.

Our results showed for the first time the role of FOH in the neutrophil recognition of fungal infection at an early stage of microbial invasion. It seems that the regulation of fungal cell morphology by FOH during the progress of fungal infection can also be sensed by the host. Neutrophils, the cells of the first line of host defense can “eavesdrop” the fungal cell communication that uses the quorum sensing molecule and then quickly migrate to the pathogen’s habitat and kill or limit its spread by using NETs. The discovery of QSMs that can activate netosis can be of great importance for the developing of effective therapy against early candidiasis.

## Figures and Tables

**Figure 1 cells-08-01611-f001:**
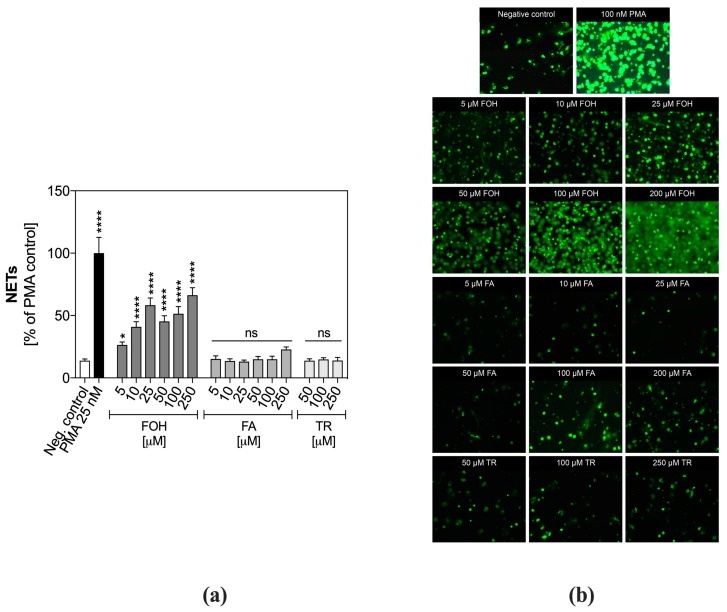
Release of neutrophil extracellular traps (NETs) by quorum sensing molecule (QSM)-treated neutrophils. Neutrophils were treated with QSMs at variable concentrations for 3 h. Unstimulated cells served as a negative control. (**a**) The released NETs were digested with micrococcal nuclease (MNase) and collected supernatants were stained with Sytox Green and the fluorescence intensity was measured. Data represent the mean fluorescence ± standard error of the mean (S.E.M.). from two replicates. ANOVA and Dunnett’s multiple comparisons post-tests were used. Asterisks denote statistical significance (*p* > 0.1234 ns, * *p* ≤ 0.0332, **** *p* < 0.0001). (**b**) The extracellular DNA was stained with Sytox Green and visualized with fluorescence microscopy.

**Figure 2 cells-08-01611-f002:**
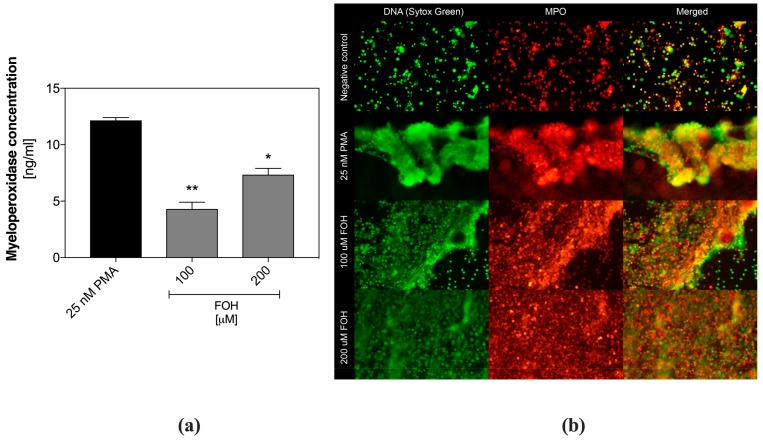
Identification of myeloperoxidase in farnesol (FOH)-treated neutrophils. Neutrophils were treated with FOH at a concentration of 100 µM and 200 µM for 3 h. Unstimulated cells served as a negative control. (**a**) The released DNA was digested with MNase, and a concentration of myeloperoxidase (MPO) was analyzed in collected supernatants using ELISA method. Data represent the mean concentration ± S.E.M. from two replicates. ANOVA and Dunnett’s multiple comparisons post-tests were used. Asterisks denote statistical significance (*p* > 0.1234 ns, * *p* ≤ 0.0332, ** *p* ≤ 0.0021). (**b**) The extracellular DNA was stained with Sytox Green, while MPO was identified with primary mouse anti-MPO antibodies and secondary Alexa Fluor 555-labeled anti-mouse antibodies. Samples were visualized with fluorescence microscopy.

**Figure 3 cells-08-01611-f003:**
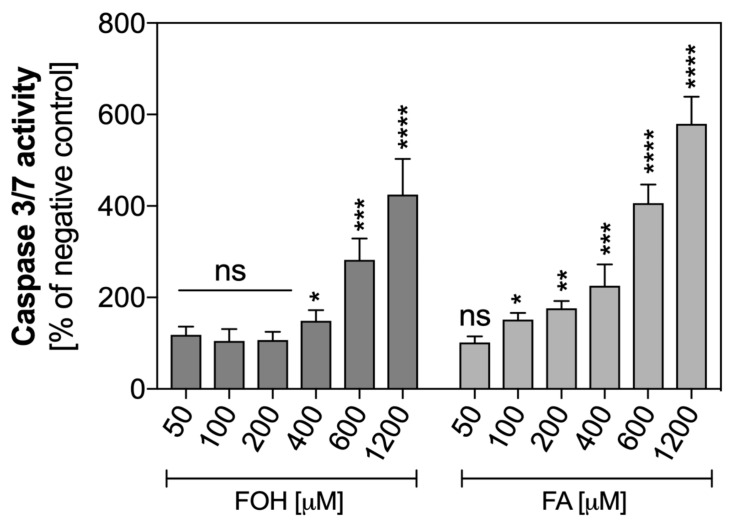
Caspase 3/7 activity in QSM-treated neutrophils. Neutrophils were treated with FOH and farnesoic acid (FA) at variable concentrations, and incubated with Caspase-Glo^®^ 3/7, followed by continuous measurement of chemiluminescence for 2 h. Data represent mean values of luminescence from two replicates ± S.E.M. ANOVA and Dunnett’s post-tests were used. Asterisks denote statistical significance (*p* > 0.1234 ns, * *p* ≤ 0.0332, ** *p* ≤ 0.0021, *** *p* ≤ 0.0002, **** *p* < 0.0001).

**Figure 4 cells-08-01611-f004:**
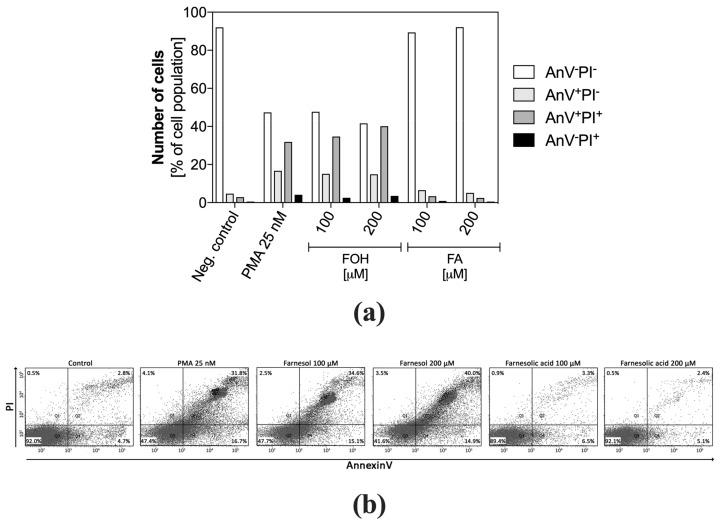
Flow cytometric analysis of neutrophil apoptosis. Neutrophils were stimulated with FOH or FA, labeled with annexin V-FITC and propidium iodide, and analyzed by flow cytometry. The results are presented as a percentage ratio of the signal detected for whole cell population and showing no cell death (PI^−^AnV^−^), early apoptosis (PI^−^An^+^) and late apoptosis (PI^+^AnV^+^). Cells stained only with propidium iodide (PI^+^AnV^−^) represent the necrotic or NET-forming cells. For each sample, data were collected for 100,000 neutrophils. (**a**) Data are presented as a part of total cell population, (**b**) the diagrams represent AnV and PI cell distribution.

**Figure 5 cells-08-01611-f005:**
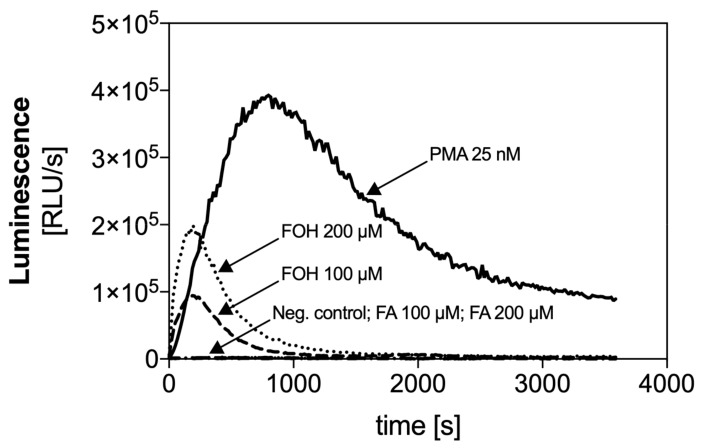
The time course of reactive oxygen species (ROS) generation by neutrophils in response to FOH and FA. ROS production by neutrophils (2 × 10^5^ cells/well) was monitored at 37 °C for 1 h by the luminol chemiluminescence method after suspension of cells in FOH or FA. In the reference samples, the neutrophils were treated with 25 nM phorbol 12-myristate 13-acetate (PMA) or left in Krebs–Ringer phosphate buffer (a negative control).

**Figure 6 cells-08-01611-f006:**
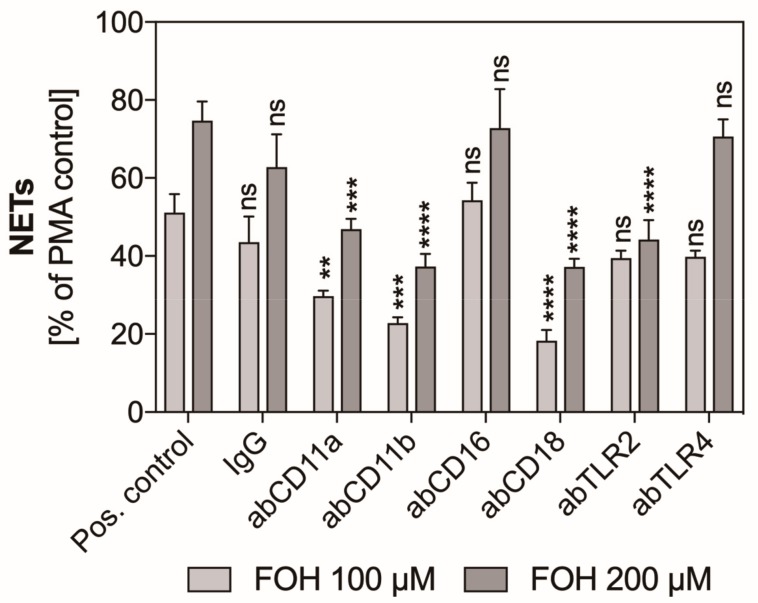
The participation of selected neutrophil receptors in FOH-activated netosis. Neutrophils (2.2 × 10^5^) were preincubated with antibodies (ab, 1 µg/mL) against the selected neutrophil receptors and then netosis was induced for 3 h by 25 nM PMA, 100 µM FOH, and 200 µM FOH. IgG antibody was used as an isotype control. Released NETs were digested with MNase, and collected supernatants were stained with Sytox Green, followed by fluorescence measurements. The results are the means of two replicates ± S.E.M., represented as a percentage relative to the PMA-treated control. ANOVA and Dunnett’s post-tests were used. Asterisks denote statistical significance (*p* > 0.1234 ns, ** *p* ≤ 0.0021, *** *p* ≤ 0.0002, **** *p* < 0.0001).

**Figure 7 cells-08-01611-f007:**
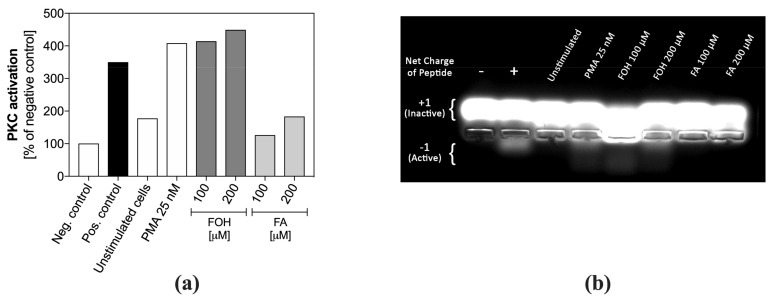
Activation of Protein Kinase C (PKC) in QSM–treated neutrophils. Neutrophils (5 × 10^5^) were stimulated with selected concentrations of FOH, FA, and 25 nM PMA for 1 h. Then, cell lysate was added to the PKC substrate solution-PepTag^®^ C1 Peptide, and after 30 min separated electrophoretically on an agarose gel. Unstimulated cells were negative cellular control, and PMA-treated cells were positive cellular control. Purified PKC enzyme was used for an assay positive control (control of C1 Peptide phosphorylation), negative control was C1 peptide without any PKC enzyme. (**a**) PKC activation as a percentage of phosphorylated C1 peptide, a substrate for PKC. The values represent the densitometrically-analyzed bands on the electrophoretic gel, assigned as “Active” in (**b**).

**Figure 8 cells-08-01611-f008:**
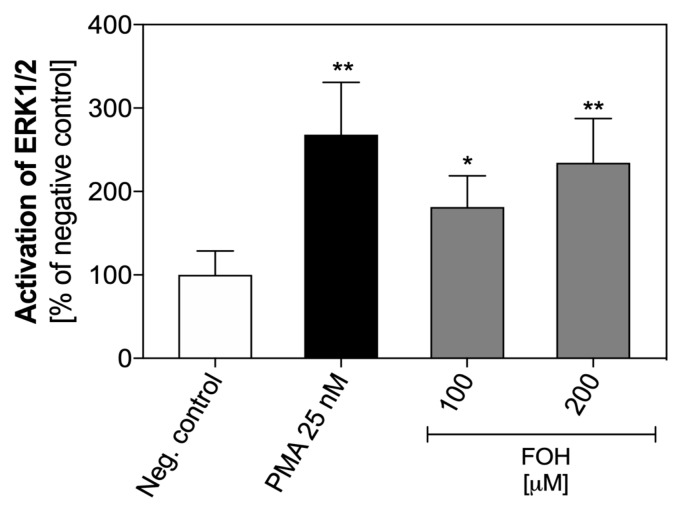
Activation of ERK1/2 in FOH-treated neutrophils. Neutrophil cells (1 × 10^6^ cells/well) were suspended in PBS in 12-well microplate and stimulated with FOH at concentrations of 100 µM and 200 µM. Unstimulated cells served as a negative control, and PMA-treated cells were a positive control. After 1 h, the cells were lysed, and the amount of total and phosphorylated ERK1/2 was quantified using ELISA Kit SimpleStep (Abcam). Data represent the mean values of absorbance ± S.E.M. from two replicates as the percentage ratio of phosphorylated ERK1/2 to total ERK1/2. ANOVA and Tukey post-tests were used. Asterisks denote statistical significance (*p* > 0.1234–ns, * *p* ≤ 0.0332, ** *p* ≤ 0.0021).

**Figure 9 cells-08-01611-f009:**
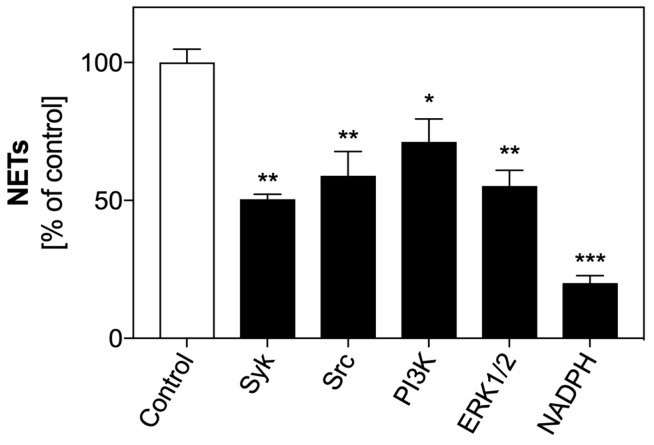
Role of selected signal mediators in activation of netosis by FOH. Neutrophils (2.2 × 10^5^) were preincubated with inhibitors of the indicated signaling mediators: Syk—30 µM piceatannol, Src—10 µM PP2, PI3K—25 µM LY29004, ERK1/2—10 µM U0126, NADPH oxidase—5 µM DPI. Cells were then incubated with FOH at 200 µM concentration for 3 h to induce netosis. Neutrophils not treated with inhibitors but stimulated with FOH served as a control. The data are presented as means ± S.E.M. from two replicates and are expressed as a percentage ratio relative to the control. ANOVA and Tukey post-tests were used. Asterisks denote statistical significance (*p* > 0.1234 ns, * *p* ≤ 0.0332, ** *p* ≤ 0.0021, *** *p* ≤ 0.0002).

**Figure 10 cells-08-01611-f010:**
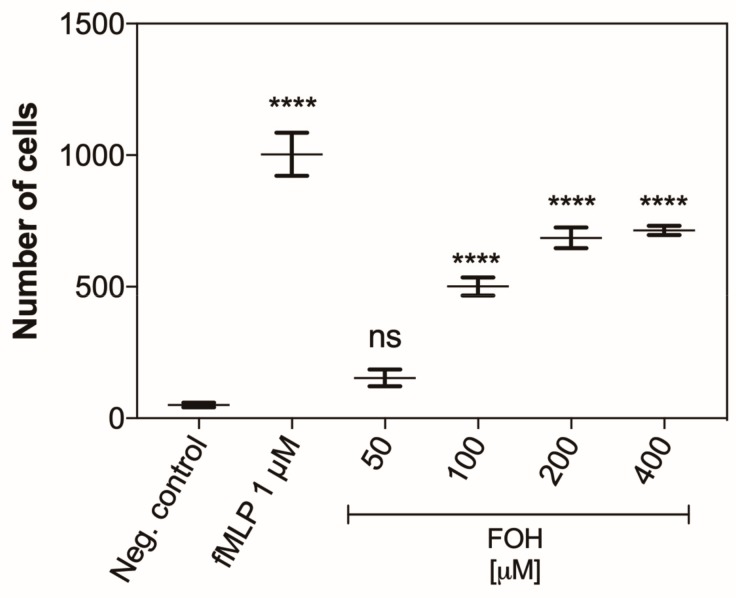
Chemotactic activity of FOH-treated neutrophils. CellTracker Red labeled neutrophil cells (1 × 10^6^ cells/well) were placed into chemotaxis chambers being in contact with different concentration of FOH or 1 µM fML (a positive control). Negative control was PBS. After 1 h of incubation, the chambers were removed, and migrated cells were counted. Data represent the mean number of cells ± S.E.M. from three replicates. ANOVA and Dunnett’s multiple comparisons post-tests were used. Asterisks denote statistical significance (*p* > 0.1234 ns, **** *p* < 0.0001).

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
