# Peer review of "Farnesol, a Quorum-Sensing Molecule of Candida albicans Triggers the Release of Neutrophil Extracellular Traps"

_cells, 2019, doi:10.3390/cells8121611_

Round 1
Reviewer 1 Report
This is an interesting article in the field of neutrophil biology and Neutrophil Extracellular Traps (NETs) formation.
In last paragraph of introduction, please write the aim of the present study. You would not describe your results (page 2, line 74-79).
The experimental protocols are poorly described and details on concentrations, identification of antibodies, and origin of experimental protocols are missing.
Sytox Green dye is used to visualize extracellular DNA. It isn't marker of NETosis. It is generally accepted that NETs markers are DNA dye like Sytox Green or DAPI in parallel with specific neutrophil markers like Neutrophil elastase or MPO.
For NETs quantifications please use MPO/DNA complex ELISA.
Page 6, line 214. Please change the title of 3.2 "Farnesol treatment of neutrophils does not lead to cell death" to "Farnesol treatment of neutrophils does not lead to cell apoptosis"
Figure 6 b. Electrophoresis analysis except for positive and negative controls of PKC should show and molecular weight standards.
Author Response
- Reviewer 1
In last paragraph of introduction, please write the aim of the present study. You would not describe your results (page 2, line 79-81).The last paragraph of the introduction has been modified, as suggested by the reviewer. The aims of the work were highlighted, while information about the obtained results was removed.
The experimental protocols are poorly described and details on concentrations, identification of antibodies, and origin of experimental protocols are missing.The experimental protocols have been corrected, and all changes made have been marked in the text. The relevant information has been completed as follow:
Line 95-97 – manufacturers names: FOH (trans,trans-Farnesol; Sigma-Aldrich, St. Louis, U.S.A.), FA (Echelon Biosciences Inc, Salt Lake City, U.S.A.) or TR (Sigma-Aldrich); Line 115 – volume: 10 µl of FOH; Line 120-121 – volume: 150 μl of RPMI-1040; Line 122 – volume: 150 μl of FOH, FA or TR; Line 122 – the name of the buffer: in RPMI-1040; Line 122-123 – concentration: cells treated with 25 nM of PMA Line 130 – volume: 50 μl micrococcal nuclease; Line 137-153 – additional description of the experimental method: Identification and quantification of myeloperoxidase; Line 157-158 – manufacturers names: TLR2, TLR4 (Invivogen, Toulouse, France), CD11a, CD11b, CD16, CD18 (BioLegend, San Diego, U.S.A.); Line 167-179 – volumes and concentrations: 5 μl of PepTag® PKC Reaction Buffer, 2 μg of PepTag® C1 Peptide, and 5 μl of sonicated PKC Activator, 10 μl of purified samples or 4 μl of Protein Kinase C at a concentration of 2.5 μg/ml; Line 182 – information about antibodies: provided by the manufacturer; Line 184 – the name of substratum: TMB (3,3',5,5'-tetramethylbenzidine); Line 189-192 – manufacturers names: (Syk inhibitor; Sigma-Aldrich), 10 μM PP2 (Src inhibitor; Calbiochem, Darmstadt, Germany), 10 μM UO126 (ERK inhibitor; Cell Signaling Technology, Beverly, USA) or 5 µM DPI (NADPH oxidase inhibitor; Sigma-Aldrich); Line 196 – information about washing: washed three times with PBS; Line 198 – concentration: fMLP (1 mM; used as a positive control); Sytox Green dye is used to visualize extracellular DNA. It isn't marker of NETosis. It is generally accepted that NETs markers are DNA dye like Sytox Green or DAPI in parallel with specific neutrophil markers like Neutrophil elastase or MPO.Of course, Sytox Green labeling is not conclusive on NETs. According to the reviewer’s suggestion, we used myeloperoxidase as a specific marker for the identification of formed NETs. The presence of MPO was confirmed using specific antibodies (Abcam, ab9535).
The supplemented results are presented in the manuscript as figure 2b (renumbering of the subsequent figures).
For NETs quantifications please use MPO/DNA complex ELISA.We used ELISA for the quantitative determination of MPO released from DNA-MPO complexes after DNA degradation with MNase. The results are shown in figure 2a.
Page 6, line 214. Please change the title of 3.2 "Farnesol treatment of neutrophils does not lead to cell death" to "Farnesol treatment of neutrophils does not lead to cell apoptosis".The change suggested by the reviewer has been introduced (line 263).
Figure 6 b. Electrophoresis analysis except for positive and negative controls of PKC should show and molecular weight standards.The electrophoretic separation of the peptides was carried out according to their charges, not molecular masses. The peptide being a substrate for PKC, possess a positive charge (+1), and the peptide being the product of active PKC gains a negative charge (-1). In Figure 7b (a new number of the figure), the description of the charge and the form of PKC (active/inactive) was added.
Reviewer 2 Report
General comments:
In this paper, Zawrotniak et al report that, among quorum-sensing molecules of Candida albicans, farnesol specifically induces Netosis. The findings are interesting and may be pathologically important. There are several concerns which should be clarified.
Specific comments:
1. Figs. 1, 2, 5, 7 and 8: Does “two replicates” mean “two experiments”? It should be noted that more than three experiments are required for appropriate statistical analysis and calculating SD. Does “three replicates” for Fig. 9 mean “three experiments”?
2. Figs. 1, 2, 5, 7, 8 and 9: Several symbols (such as * and ****) are used. Clarify the p value of each symbol.
3. Fig. 1B: The images of cells treated with appropriate concentrations of FA, TR and PMA (positive control) should be presented.
4. Figs. 3 and 6: Clarify the number of experiments. How many experiments were done?
5. Fig. 5: PMA-induced Netosis is significantly enhanced by abCD18. Some discussion is required.
6. Others:
Line 136: Dectin-1 and CD14 should be deleted, since there are no data for them.
Line 174: fMLP (100 microM) may be fMLP (1 microM).
Line 199: C. albicans-relased should be C. albicans-released.
Fig. 6, Figure legend: Underlines may be deleted.
Author Response
- Reviewer 2
1, 2, 5, 7 and 8: Does “two replicates” mean “two experiments”? It should be noted that more than three experiments are required for appropriate statistical analysis and calculating SD. Does “three replicates” for Fig. 9 mean “three experiments”?Each of the experiments was repeated at least three times, obtaining consistent results. Two replicates were performed in each experiment. The graphs show the results of a single representative experiment.
We agree with the reviewer that the presentation of the results with S.D. was not the right choice. All charts have been changed, and they show the results with S.E.M.
1, 2, 5, 7, 8 and 9: Several symbols (such as * and ****) are used. Clarify the p value of each symbol.Information about the symbols used for the determination of the statistical significance (p-value) has been implemented in all legends of the figures.
1B: The images of cells treated with appropriate concentrations of FA, TR and PMA (positive control) should be presented.Figure 1b has been supplemented with microscopic images obtained for neutrophils stimulated with FA and TR.
3 and 6: Clarify the number of experiments. How many experiments were done?Each of the experiments was repeated at least three times, obtaining consistent results. Two replicates were performed in each experiment. The graphs show the results of a single representative experiment, with S.E.M.
5: PMA-induced Netosis is significantly enhanced by abCD18. Some discussion is required.The reviewer's comments suggested the revising of figure 6 (the new number of the figure) and changing the way of the result presentation. In the original version, all neutrophil responses were presented as a percentage of the fluorescence intensity of stained DNA, released by neutrophils stimulated with PMA (positive control).
Such a presentation did not take into account the non-specific variability of neutrophil responses to PMA treatment, in the presence of antibodies. Therefore, the values showed in the new graph represent the percentage of neutrophil response to FOH relative to the response to PMA, in the presence of tested antibodies.
The unexpected but statistically significant difference in neutrophil response to PMA, identified in the presence of anti-CD18 antibodies was the results of incorrect conversion of the fluorescence measurement data.
Line 136: Dectin-1 and CD14 should be deleted, since there are no data for them.The not relevant information has been removed.
Line 174: fMLP (100 microM) may be fMLP (1 microM).The information on the proper concentration of fMLP used in the experiments was introduced.
Line 199: C. albicans-relased should be C. albicans-released.The letter error has been corrected.
6, Figure legend: Underlines may be deleted.Underlines have been removed.
Reviewer 3 Report
This article “Farnesol, a quorum-sensing molecule of Candida albicans triggers the release of neutrophil extracellular traps” belongs to the field of the host's innate immune responses to pathogens and is focused at the effects of quorum-sensing molecules produced by Candida albicans on NETs formation in neutrophils.
I guess that farnesol used in this study has the same formula as farnesol (FOH) from the oils of diverse plants ((2E,6E)-3,7,11-trimethyldodeca-2,6,10-trien-1-ol), MW=222 (https://doi.org/10.1016/j.phrs.2019.104504). Farnesol has three carbon-carbon double bonds and exists in four isomers. It would be good to give some comments about the structure of farnesol used in this study, is it the mixture of isomers? In the Introduction (lines 60-62) phagocytosis and NET release are introduced as cellular responses of neutrophils to C. albicans. Yes, NETs, like phagocytosis, is a mechanism to eliminate pathogens, but NETosis is a way for neutrophil death. There is no information in the introduction that the formation of NET is associated with the destruction of the plasma membrane and the release of chromatin, which leads to the formation of NET. Live neutrophils create a cytoneme to catch and kill pathogens extracellularly (DOI: 10.1016/j.yexcr.2004.12.005; 10.1016/j.bbagen.2012.06.016; DOI: 10.1038/embor.2013.89). Lines 139-164. It is not clear, if the cells treated by 100-200 µM FOH, with disturbed membrane integrity, keep their proteins inside the cell. Did you check the medium before washing for presence of PKC and ERK1/2? Figure 1. Detection of NETs using only the DNA-binding dye Sytox Green is not sufficient for expert level work. Sytox Green is impermeant to live cells. At the same time, the nature of the glow indicates both extracellular and intracellular staining. Cells are dead at 100-200 µM? In dead cells intracellular DNA may interfere assay. Figure 2. Caspase 3/7 activity assay is a marker of the way to cell death by apoptosis. Formation of NETs is a mechanism of neutrophil death, which is different from apoptosis and necrosis, it does not require caspases and is not accompanied by DNA fragmentation (DOI: 10.1038/nrmicro1710). So, these data mean that farnesol treatment of neutrophils does not lead to cell death by apoptosis and cannot be interpreted as Viability assay. Figure 3 shows that after 1-hour incubation with 100-200 µM FOH about 40 % of neutrophils are PI-positive, i.e. they lose membrane integrity. For discussing neutrophil apoptosis (Lines 245-255), these results need to be supported with another assay; e.g. TUNEL staining or equivalent.All this in the complex does not allow a detailed analysis of the results, since the adequacy of the methodology used raises doubts.
Author Response
Please, see the attachment

Round 2
Reviewer 1 Report
All my earlier comments have been sufficiently addressed. I recommend publication of this manuscript
Reviewer 2 Report
The manuscript is appropriately revised.
Reviewer 3 Report
The authors took into account all the comments and made all the necessary corrections